# Mental and Physical Well-Being of Partners of People Living with Chronic Pain: A Narrative Review

**DOI:** 10.3390/ijerph22020205

**Published:** 2025-01-31

**Authors:** Toby R. O. Newton-John, Shari Cave, Debbie J. Bean

**Affiliations:** 1Graduate School of Health, University of Technology Sydney, Sydney, NSW 2007, Australia; 2Department of Anaesthesiology and Perioperative Medicine, Health New Zealand Waitematā, Auckland 0620, New Zealand; shari.cave@waitematadhb.govt.nz; 3Centre for Person Centred Research, School of Clinical Sciences, Auckland University of Technology, Auckland 0620, New Zealand; debbie.bean@aut.ac.nz

**Keywords:** chronic pain, partner, quality of life, spouse, coping

## Abstract

This narrative review aims to explore the mental and physical well-being of partners of individuals living with chronic pain. Chronic pain not only affects those who suffer from it, but also significantly impacts the lives of their partners; however, the impacts on partners are not well recognised, despite extensive evidence indicating that their quality of life can be equally affected. This review synthesises current literature to identify the psychological and physical challenges faced by these partners, including increased stress, anxiety, depression, and the potential for developing chronic health conditions themselves. A search of Medline for “chronic pain” and “partner/spouse” from January 1990 to the present was performed, and relevant articles were selected for review. The main findings were that while partners often experience a range of negative physical and psychosocial impacts on their quality of life, dyadic coping strategies can mitigate these effects. This review underscores the importance of future research to develop targeted interventions that address the unique needs of this population, promoting better health outcomes and fostering resilience in the face of chronic pain.

## 1. Introduction

An important element of primary romantic relationships is the idea that couples undertake to support each other, despite the challenges that they may face in life. Couples in committed relationships accept the need to support each other—whether they are physically and emotionally well, or they are not. This means that when one member of a couple falls ill, there is an inevitable impact on the other member of the dyad—and that couples need to understand they are accepting that those impacts may be part of a future relationship together. As has been noted by Adelman and others, being a carer in support of someone living with a chronic illness can bring with it a host of physical, psychosocial, financial, and other stressors that significantly add to the strain for the person providing care [1]. While this carer burden might be most evident where the chronic illness requires obvious physical assistance, as, for example, in cerebral palsy or spinal cord injury, negative impacts on the carer are not limited to these clinical groups. Partners of people living with chronic pain can be similarly affected, and this is the topic of the present review.

### 1.1. Chronic Pain—Prevalence and Impacts on the Individual

Acute pain (defined as any short-term pain lasting less than three months) and sub-acute pain (defined as pain lasting between six weeks and three months) are virtually universal human experiences. They are also typically associated with defined tissue damage or observable pathological processes [2,3,4]. Chronic pain, defined as pain that persists on a more or less daily basis for longer than three months, is a surprisingly pervasive and debilitating condition that the World Health Organization has recently estimated to affect approximately 20% of adults worldwide [5]. It has also been estimated that chronically painful conditions result in severe limitations in daily activities and quality of life for upwards of 8% of adults in the USA alone—more than 17 million people [6]—with lower back pain now the primary cause of disability worldwide out of 156 conditions assessed [7]. Unlike in acute or subacute pain states, in many cases, chronic pain develops slowly over time, such as with arthritis or lower back pain conditions. Chronic pain can also develop following a surgical intervention for relatively common reasons, such as a mastectomy, joint replacement, or appendectomy; it is estimated that 10–50% of surgeries result in a persistent pain state [2,3]. Chronic pain can also be the result of changes to the central nervous system; hence, terms such as ‘central sensitization’ and ‘hypersensitivity’ are often used to denote these chronic pain states which do not involve directly observable tissue damage.

It is no surprise then that given such widespread prevalence, there is a massive public health and economic burden associated with chronic pain conditions. Data indicate that the cost of diagnosing and managing pain already exceeds the costs associated with heart disease, cancer and diabetes, and the financial impacts will only increase, as the prevalence is expected to increase [4]. However, the costs associated with chronic pain are not solely related to finance and economics. There are often considerable psychological difficulties for a person with chronic pain (PWCP) that are associated with living with intractable chronic pain over many months and years, despite many attempts to find a cure via treatments and various interventions.

Depression is a common concomitant when pain limits the pursuit of activities that used to provide purpose and meaning in life [8], such as engaging in family life, hobbies, and social interactions. The effects of low mood can be compounded by problems with sleep [9], making each worse in the presence of the other [10]. For a PWCP, anger, frustration and a sense of injustice are also highly prevalent [11], particularly when the onset of the pain condition is related to an injury or accident that is perceived to have been the fault of another individual [12]. Reciprocal interactions between all of these negative emotional states and the experience of pain are well documented [4], such that a mutual maintenance cycle can be established. Mutual maintenance refers to the reciprocal and reinforcing relationship between factors common to two conditions [13]. In this case, being in pain has a negative effect on mood—and, as mood decreases, it leads to increased sensitivity to pain [14]. Hence, each factor perpetuates or exacerbates the other, creating a cycle that makes recovery from either condition more challenging.

Finally, chronic pain often has significant impacts upon work and occupational functioning [15], not solely due to the physical limitations associated with prolonged sitting, standing, lifting and so on, but also due to cognitive effects of reduced concentration and memory impairment [4,15] that frequently accompany persistent pain. There are obvious financial implications associated with loss of work, but from a psychological perspective it is more than that—the loss of identity, of social connection, of having a sense of value and meaning in the world—all of these factors can be part of being employed [16]. Chronic pain, and chronic lower back pain in particular, is a major obstacle to occupational functioning. According to Schofield and colleagues, back pain and arthritis are the leading chronic conditions associated with premature exits from the labour force [17].

### 1.2. Treatment of Chronic Pain

The management of chronic pain is challenging and complex, ideally involving an interdisciplinary and multimodal approach that addresses both the biological and psychosocial aspects of the condition [18]. Chronic pain treatments typically begin with intervention therapies and may include spinal cord stimulation, injection treatments of steroids and anaesthetics, or nerve ablation techniques [4]. Patients may be trialled on a range of different pharmacological agents, either singularly or in combination, in an attempt to balance the benefits of reducing pain with the costs of the inevitable side effects. Medications for chronic pain include analgesics, non-steroidal anti-inflammatories, anticonvulsants, antidepressants, and muscle relaxants. Physical therapies for chronic pain are also commonly administered, and range from massage and traction to more active patient involvement methods such as stretching and exercise-based treatments [19]. The therapeutic value of providing patients with information about the neuroscience of chronic pain has also been evaluated, both in conjunction with physiotherapy and as a stand-alone intervention [19].

However, the effectiveness and safety of these treatments vary widely and depend on individual factors such as the type, intensity, duration, and cause of pain, as well as the patient’s preferences, expectations, and adherence [18,20]. Moreover, some treatments, especially opioids, have serious adverse effects and risks of misuse, addiction, and overdose, leading to the opioid crisis that has claimed thousands of lives and harmed countless others [20,21,22].

Unfortunately, while there are many different types of treatment for chronic pain as have been outlined above, none of them are curative and sustained pain relief is a rarity for PWCP [18]. Chronic pain is just that, a long-term, persistent condition, despite the best efforts of those who treat it and those who seek relief from it.

### 1.3. Beyond the Individual

In contemporary Western society, more than 60% of adults either marry or cohabit [23]. Given all of the negative impacts that have been documented above regarding chronic pain’s effects upon the individual, we can turn our attention now to the focus of this article—the impact of chronic pain upon the partner. We have chosen to use the term ‘partner’ as synonymous with ‘significant other’, ‘spouse’, ‘romantic other’ and any other term which refers to a primary, cohabiting relationship.

Why is it important to explore the mental health of the chronic pain partner as well as the PWCP? Often, partners are not included when considering the ways in which living with pain influences day-to-day functioning, which is a significant oversight considering that dyadic adjustment is one of the most important contributors to mental health [24]. In fact, dyadic adjustment, or the extent to which partners adapt to one another, manage conflict, and maintain satisfaction and cohesion in their relationship, is a stronger predictor of mental health than the other way around [25]. This underlines the influence that relationship factors have in terms of affecting individual well-being [26,27].

The importance of the partner is also critical when considering the reciprocity inherent in close relationships. As will be shown, many studies have demonstrated how the well-being of the PWCP can impact the partner, but the influence is bidirectional.

## 2. Impact of Chronic Pain on the Partner

As described above, extensive research has documented the effects of chronic pain on the quality of life and well-being of the PWCP, however relatively less is known about the effects on the partner of the PWCP. Therefore, a search of the Medline database was performed by the first author using the Medical Subject Headings “chronic pain”, “partner” and “spouse” from January 1990 to 1 July 2024. Articles considered relevant for this review were selected and assessed, based on inclusion criteria relating to studies in which results (either qualitative or quantitative) reported on the partner’s experience of chronic pain. Studies relating to cancer pain or other terminal illnesses were not included. All authors were involved in determining which articles would be included in the review. The research that does exist (described below) demonstrates that partners experience a range of practical demands, relationship changes, and challenges to their psychological and physical well-being. This provides compelling insight into the importance of supporting both partners when one (or both) have chronic pain.

### 2.1. Meeting Practical Demands

Partners of PWCP may experience significant disruption to daily routines, activities, and quality of life. Both qualitative and quantitative studies have documented how partners pick up additional household tasks and parenting responsibilities when the PWCP can no longer meet these demands [28,29,30,31,32,33], with more than 50% of spouses in one study reporting moderate to severe problems with duties around the house [34]. Partners also engage in caregiving roles and ‘illness work’, including attending appointments, helping with treatment decision making, and contributing to managing the medication, symptoms or activities of the PWCP [31,32,35].

The impact of chronic pain on occupational functioning for PWCP was noted previously, but partners may also be affected here. If the PWCP is unable to work, partners may need to return to work, increase their work hours or seek different or higher paid employment due to financial pressure [29,30,32,36]. If the PWCP requires caregiving or support at home, partners may take more flexible work, use leave, work from home, or find employment closer to home [28,32,33,36]. One study found that 27% of spouses had a change in their own work status as a result of their partner’s pain [29]. Financial limitations may also result from the loss of income if the PWCP is not working or due to healthcare costs [30,32,33,34,37] and 35% of the spouses of people with fibromyalgia reported a loss of household income due to the pain condition [38]. Such challenges may affect quality of life, and one study of the overall impact of a partner’s illness found that partners of PWCP scored higher on a measure of impact than partners of people with various types of cancer [34].

### 2.2. Relationship Changes

Consistently, studies show that chronic pain influences the nature of a relationship. Sex and intimacy are frequently impacted, not only by pain itself, but also due to accompanying fatigue, medication, mood changes or fear [31], and partners describe how their role changes from one of being a lover to being a caregiver [28,30,32,33,36,37,39]. For example, in one qualitative study of 26 partners of people living with chronic back pain receiving treatment at an outpatient rehabilitation centre, one woman said “I love him and need him as a husband. But I feel like his nurse” [35]. Male partners of women with fibromyalgia described “facing a new sex life”, which involved resisting the loss of the couple’s sexuality and finding ways to enhance intimacy and increase partner desire [39]. In contrast, partners of people living with migraine described having to embrace the role as caregiver and find satisfaction in caring for their spouse [37].

In addition to impacts on intimacy, chronic pain can cause relationship tension and dissatisfaction. Partners of PWCP in one study described the difficulties of living with a person who may be frustrated or angry, and the need to distance themselves from the PWCP as a protective strategy [30]. Partners may feel it is inappropriate to ask for support from the PWCP because of the inherent stress of living with pain on a daily basis and therefore deliberately not communicate their own needs and wishes to the patient, resulting in a change to the give-and-take nature of the relationship [28,31,37]. Marital dissatisfaction or relationship burden were reported by approximately 50% of partners of PWCP [29,40] indicating that strain and tension occur for many couples (though this is not always the case).

Several studies described how, for some couples, the communication, dependency, team-work and compassion required when facing chronic pain together had brought them closer together, and thus chronic pain can lead to both disconnection or connection [28,36,37]. A recent qualitative evidence synthesis of 15 primary studies described how in light of the challenges of living with chronic pain, partners may re-evaluate their commitment to the relationship [28]. A negative re-evaluation could exacerbate dissatisfaction whereas a positive re-evaluation can lead to an enhanced level of commitment. This review found that factors such as a lack of open communication, or heightened expressions of anger and fluctuating moods, were associated with greater relationship strain for both patient and partner. Alternatively, expressions of care, compassion, and valuing time together were the variables related to positive relationship adjustment [28]. Thus, the effects of chronic pain on a couple’s relationship are highly dependent on the ways in which couples are able to negotiate the stresses of the condition together, as the evidence shows that while chronic pain is undeniably stressful, a deterioration in the quality of the relationship is not an inevitable outcome if the previously mentioned positive relationship variables can be realised.

In an attempt to understand how it is that, for some relationships, the development of a chronic pain condition can lead to heightened stress and dissatisfaction, and for other couples it leads to greater closeness and intimacy, attachment theory is increasingly being considered [41,42]. Formed as a result of early life experiences of caregiving, adult attachment styles are broadly categorised as secure or insecure [41]. Adults with a secure attachment style are comfortable with having close relationships, but are not fearful of being on their own; by contrast, insecurely attached adults may either avoid intimacy at all costs (avoidant attachment) or cling desperately to a relationship despite obvious negative consequences (anxious attachment). As will be discussed later in this review, the role that attachment style may play in influencing a couple’s adjustment to chronic pain is of increasing importance.

### 2.3. Psychological Well-Being

Supporting a partner with chronic pain and its associated demands may cause emotional distress as couples navigate complex changes and losses. First, partners of PWCP may experience social isolation and loneliness [28,38]. Partners describe being unable to commit to social events themselves or as a couple, and report that friendships are affected [33,36,37]. Friends or relatives may distance themselves from the couple due to judgement or social stigma regarding the chronic pain [30,38]. Partners may also receive less support from within their relationship; for example, they may sacrifice their own emotional needs to prioritise supporting the PWCP, or even conceal their own negative thoughts and emotions to stay positive and supportive [28,32].

Second, partners also describe heightened experiences of negative emotions as they manage and come to terms with living with PWCP and the loss and grief this may include. Partners in several studies discussed “endless uncertainty”, fear or worry they experience due to the pain and the effect it may have on the couple’s future life [28,30,31,36]. They may experience a sense of helplessness at seeing the PWCP suffering and not being able to help [43], as well as a desire to protect and shield their spouse [36,44]. Partners report experiencing frustration or anger as a result of the pain [31,43], or directed towards the PWCP [30]. This negative emotion could then be compounded by a sense of guilt or self-blame about the resentment they experience [28,30].

Third, the partners of PWCP may be at higher risk of depression. Higher rates of depression have been reported in partners of PWCP relative to community samples [40] and partners of healthy individuals [38]. However, rates of depression remain lower in partners relative to the PWCP [40], and in one study elevated levels of depression were attributable to chronic pain symptoms in the partners themselves, noting that many partners also experience pain [45]. Several studies suggest that these negative mood states may be directly related to the chronic pain condition, as partner mood or distress has been linked to the disease severity [46], or the pain intensity of the PWCP [47], or the couple’s approach to coping with pain [48].

### 2.4. Physical Well-Being

Relatively little research has assessed the physical health of partners of PWCP. Although evidence shows that caregiving for a spouse, in general, is associated with increased cardiovascular risk [49], supporting a partner with chronic pain specifically may be different from caring for a partner with one of the degenerative or terminal conditions typically included in caregiver studies. In one study which did explore cardiovascular health, rates of high blood pressure, diabetes and heart disease were similar between the partners of people with fibromyalgia (a common chronic pain condition) and the partners of healthy people [38], suggesting cardiovascular risk may not be increased in chronic pain states. Interestingly, it has been found that partners of PWCP display heightened autonomic arousal when observing their spouse displaying pain compared to observing their partner with a neutral facial expression [50], but whether this arousal ends up affecting health in the longer term is unclear.

Interestingly, partners of PWCP may experience elevated levels of pain themselves. Partners of PWCP reported more pain symptoms relative to spouses of people with diabetes [40], and one study found that nearly half of partners of PWCP report pain problems such as knee pain, lower back pain or osteoarthritis themselves [45]. Partners may also experience sleep problems, with one diary study demonstrating that patients’ daily knee pain intensity predicted partners’ sleep quality [51].

## 3. Interventions for Couples with Chronic Pain

Chronic pain interventions have typically been delivered at the individual level of the person with chronic pain (PWCP) and often do not address the social context in which the PWCP exists. Intimate partner relationships and those in these relationships are an important part of a person’s social context and are frequently impacted by the presence of chronic pain [29], while also impacting chronic pain [52]. In recognising the importance of the partner and relational factors, work has been carried out to explore the impact of including partners of PWCP in chronic pain interventions. Smith and colleagues recently carried out a systematic review of 23 studies assessing couples’ interventions for chronic pain [53]. Educating couples about chronic pain was the most common strategy (83% of studies), with teaching relaxation or meditation skills to both patient and partner being the next most common type of intervention (48% of studies). In terms of outcome variables assessed, pain intensity was assessed in 15/23 studies (65%) and either a measure of physical function and/or psychological well-being was assessed in 70% of studies [53].

However, when comparing couples-based interventions with interventions delivered to the patient only, or with waiting list control groups, this review revealed only modest benefits for couples-based treatments. Only two studies reported significant reductions in patients’ pain intensity [54,55], while six studies reported improvements in patients’ pain-related outcomes, e.g., depression, pain coping, and functioning [54,56,57,58,59,60]. These results suggest that the overall impact of couples’ interventions for chronic pain may be somewhat limited.

One explanation for the modest benefits observed in these interventions is that, in most cases, the PWCP remains the primary focus of treatment, while the partner is included in a supportive or coaching role, such as in partner-assisted coping skills training [53,61]. Notably, only a small subset of studies have measured outcomes for the partner [58,59,60,62,63,64], and among those that have, only a few report significant improvements [65,66,67]. Thus, it seems that the partner’s well-being and the relationship are not being targeted in these interventions, and hence partners are not benefitting. Given the importance of partner well-being and relationship factors in the context of chronic pain, a more relational approach—one that targets both the partner and the relationship itself—could help mitigate the interpersonal strain and the individual distress associated with chronic pain.

Research has begun to explore relational factors in couples with chronic pain, in particular in relation to dyadic coping. Dyadic coping refers to the ways in which stress is managed at the level of the couple, rather than at an individual level [68]. Recent work examining the role of dyadic coping in couples with chronic pain found that negative dyadic coping was associated with poorer relationship quality, as well as higher levels of depression, anxiety, and stress for both the PWCP and the partner [48]. In contrast, positive dyadic coping was linked to better relationship quality and reduced depressive symptoms in the partner [48]. While research in this area is relatively new, some promising findings have emerged from interventions that directly target partner well-being and relational factors. For example, a motivational interviewing assessment intervention led to significant reductions in pain severity (self- and partner-reported), negative mood, and increased marital satisfaction compared to an educational control [69]. Similarly, an intervention focusing on psychological and relational flexibility in couples showed reductions in pain and pain-related interference for the PWCP, along with increased social engagement for the partner [70]. Both members of the couple also reported enhanced emotional intimacy, physical affection, and overall relationship satisfaction [70]. Such work highlights the value of addressing relationship factors for chronic pain couples, not just the needs of the PWCP, to provide meaningful benefits to both partners. However, as is the case for much of this clinical area, more research is needed to fully explore the potential benefits of these approaches.

## 4. Conclusions

There is substantial literature demonstrating a range of negative impacts of living with persistent pain for individuals. However, this review has highlighted that the partners of PWCP can also experience a range of physical and mental consequences which are much less frequently taken into account. Research has shown that chronic pain-related effects, such as mobility restriction, depression and anxiety, sleep disturbance, and work impairments do not only affect the PWCP; they are also experienced by the partner. Relationship roles and satisfaction, sex and intimacy are all affected. Similarly, research has shown that coping with the negative effects of chronic pain can occur at the level of the dyad and not just for the individual with pain. Dyadic coping improves outcomes for both members of the couple. Examples of dyadic coping include the partner playing an active role in decision making about pain treatment, the PWCP not always prioritising their pain management requirements and instead considering the partner’s needs and wishes, and finally when communication between the couple is empathic and supportive.

This review has shown that treatments for chronic pain couples have historically not taken the partner’s health into account, despite the evidence that many partners are negatively affected by the consequences of pain—which can in turn negatively impact the PWCP. The development of future interventions for chronic pain couples will need to consider issues of diversity, the role of attachment style in both members of the dyad, and a move away from a partner-as-coach orientation towards a relational approach which addresses the needs of both members of the couple.

## 5. Future Directions

Having outlined the current status of research and the various ways in which chronic pain can impact the well-being of both the PWCP and the partner, it is possible to consider the areas that remain inadequately addressed in this research area. First among them is the fact that to a large extent, the literature on couples is based on cross sectional observations of white, middle class, heterosexual couples in long term, stable relationships. This research bias has meant that little is known about the impact of pain on relationships across time, as well as the processes involved in relationship adjustment (or failure to adjust) over time. It also means that little is known about the unique experiences of couples from non-European cultures, including those who form parts of racial minority groups in Western countries, as well as those from lower socio-economic backgrounds. Given that ethnicity and socio-economic status are known factors associated with chronic pain [71,72], this is a significant omission in the literature.

Knowledge in this area is also limited by a lack of longitudinal research. For example, theoretical models relating to solicitousness [73] hypothesise that partner responses to patient expressions of pain behaviour exert an influence on patient functioning when delivered repeatedly over time [74]. However, in order to confirm this theoretical proposition, longitudinal research designs are required to assess the effect of ongoing partner behaviours—but these are very much lacking in this area. As was previously noted, there is a further research limitation in that relatively few studies in this area have measured outcomes for both patients and their partners [58,59,60,62,63,64], despite the emphasis on relationship dynamics.

Another area that has shown promise but is yet to be fully explored is the role of attachment theory and chronic pain couples’ adjustment [41,42]. Insecure attachment has been identified as a reliable predictor of poor coping with both acute and chronic pain at the individual level. Attachment theory further suggests that the dyadic interactions of anxiously attached and avoidantly attached individuals will be quite different [42], and hence will require different kinds of intervention. However, we do not yet know how these differing types of insecure attachment exert their influence on patient communication styles in their intimate relationships, and nor is there research exploring how the partner’s attachment style interacts with that of the patient when it comes to issues such as spousal solicitousness [73,74]. There is substantial evidence that partners can inadvertently reinforce patient communications or expressions of pain over time [50,73,74,75]; however, this does not impact all couples in the same way. Some PWCP report being highly desirous of their partner’s attentive responses when they are in pain, hence the potential for pain behaviours to be reinforced; others report a preference for being ‘left to get on with it’ [44]—the attachment styles of both patient and partner are likely to play a significant role in these interactions.

And finally, in relation to treatment itself, the next iteration of couples-based chronic pain interventions will likely involve a greater focus on the communication of empathy and less of a focus on applying concepts from operant behavioural theory [75,76]. As was noted previously, many of the early chronic pain couple’s treatment studies were primarily directed at improving patient functioning using a partner-as-coach/guide approach, rather than testing treatments aiming to improve the functioning of the relationship. Partner outcomes were often not even assessed, as these interventions were not intended to address relationship factors (as relationship factors were not considered integral to optimal patient outcomes). Rather, they were designed to teach the partner how to notice and reinforce the PWCP’s well behaviours and avoid reinforcing pain behaviours [63,67]. While these interventions had solid theoretical foundations, the evidence for their effectiveness is relatively modest. As such, there is a need for future treatment outcome studies to explore the role of empathic communication in pain couples’ interactions, rather than persist with narrow behavioural conceptualizations of dyadic exchanges. Understanding the mechanisms by which pain behaviour expression and solicitous responding build empathy [68,75,76] is likely to lead to interventions which can significantly improve the quality of life for both patient and partner.

## 6. Summary

By definition, chronic pain is a long-term health condition, and there is now a substantial amount of evidence to show that it often has a wide range of negative impacts for individuals living with pain, as well as for their partners. When considering the next steps in the field, we hope that this review encourages clinicians working with chronic pain patients to routinely ask about relationship changes and partner coping when carrying out an initial assessment with a patient. This information can be valuable when formulating as to the causes and/or maintaining factors in a patient’s presentation. As we have shown, principles from attachment theory can be usefully applied to assist in interpreting patient and partner behaviour in the context of a chronic pain condition. With regard to research, we have argued that the next major progressions in this area will come from research design improvements (greater participant diversity, more longitudinal assessment, regular inclusion of partner outcomes), and from an updating of the treatments themselves. With a greater emphasis on enhancing dyadic coping strategies and the optimisation of couple communications, we look forward to seeing a significant reduction in the psychosocial burden of chronic pain for both patients and their partners.

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
