# Peer review of "Mental and Physical Well-Being of Partners of People Living with Chronic Pain: A Narrative Review"

_ijerph, 2025, doi:10.3390/ijerph22020205_

Round 1
Reviewer 1 Report
Comments and Suggestions for Authors
This review was well written and covered the main areas of interest in the field of carers health and wellbeing. However no methodology was given for this narrative review. A scoping review may have been a more rigorous choice of literature review method. It was hard to know if the chosen literature was the result of a search or random. Keywords & databases searched would be minimum useful information.
Whilst well written I'm afraid I didn't feel this adds much new knowledge to the field.
A more general overview of what is known about carers health issues would have been welcome before honing in on partners of people with chronic pain.
Partners as a term is an interesting choice.... you didn't seem to include adult children of people with pain which make up a large proportion of family carers. Perhaps consider care supporters, or care p could artners ? Even better, you engage in consultation with people with chronic pain to confirm the best term to use.
In the conclusion (recommendations) you introduce attachment theory. This was a large new area to introduce so late in the paper. Recommend incorporating this earlier.
Overall well written but lack of methodology and evidence of literature search rigour left me unsure what original knowledge this adds to what is already known.
Reviewer 2 Report
Comments and Suggestions for Authors
The introduction starts with an, in my opinion, old school way of thinking. Its written very romantic but not true anymore. Society changed and a lot of married couples are getting divorced nowadays. This strong believe in marriage and supporting each other for the rest of their lifes does not exist anymore, like it did 50 years ago. That's why I don't think the introduction is well written. Also, love and supporting each other does not depend on marriage. There are a lot couples not married, but still supporting each other. Please overthink your introduction!
Line 39: upwards of ---> I think you mean: for up to
Line 55-57: Be careful with depression and chronic pain. It clearly coexists, but one still does not know, if the chronic pain came first or the depression came first and the chronic pain being a symptom of depression.
Line 70: you mean unemployed, right? not employed.
Line 88: Is there a reference saying that sustained painrelief is rare? Or is it your opinion?
Line 204-205: "Partners may also experience sleep problems, with one diary study demonstrating that patients’ daily knee pain intensity predicted partner sleep quality (40)." Are there any ideas of the cause for this? This seems very odd.
Line 209: "to" needs to be deleted
Line 249: "...though more research is needed..." is the start of a new sentence.
Line 253: "romantic partners..." please delete romnatic. Partners always refer to the spouse or cohabitee/significant other, whether romantic or not ;)
Comments on the Quality of English Language
There are some issues as described above.
Reviewer 3 Report
Comments and Suggestions for Authors
1- Abstract:
- the reason for the research problem should be explained.
- the main results should be highlighted
2- Introduction:
- As the authors rightly state, chronic pain is pain that lasts more than 3 months, but it should be better explained and differentiated from acute or subacute pain in terms of sudden or delayed onset.
- The different approaches to chronic pain management should be explained in more depth from the conservative point of view and the practical participation such as pain education and physical exercise.
- Refer further, with other studies in section 1.3.
- Section 3: The types of interventions and the variables analyzed at both the physical and mental levels should be clearer and more detailed.
- Have the authors only analyzed whether the person suffering from chronic pain lives with his or her partner? what about children? another relative at home? relationship time? could it have an influence?
- add some more research limitations
Reviewer 4 Report
Comments and Suggestions for Authors
Thank you for the opportunity to review this manuscript. Overall, the manuscript is very interesting and has potential, but there seems to be some gaps in research presented and conclusions drawn by the author. There are times in which it is difficult to determine the difference between the authors' opinion and cited research (Example: Line 198-200) throughout the manuscript. It may also be helpful to consider separating sentences with multiple conclusions (Example: Lines 129-131, Lines 137-139) in order to improve readability throughout.
Lastly, the manuscript is written (at times) very informally and would benefit from some changes in order to improve this. For example:
1. Frequent use of parentheses makes the article seem less formal (Example: Lines 44-45)
2. contraction use
3. Use of single vs double quotation marks is confusing. Does one indicate quote from article vs. author’s language? ( Example: Line 63 ‘mutual maintenance’ vs. line 83 “opioid crisis”)
Other specific suggestions for refining the manuscript include:
· Lines 44-45: “mastectomy, joint replacement, or appendectomy” listed as common surgeries. Are these known as common surgeries? Where does that fact come from?
· Lines 52-55: Sentence starting with “Living with intractable chronic pain…” – Run on makes it difficult to understand this sentence
· Line 63: ‘mutual maintenance’ cycle – Define. Also is this a term created by author or is this a term used in literature?
· Section 1.2. Treatment of chronic pain – Reflect on tone
o Example: Line 82: “so-called ‘opioid crisis’” – sounds like the authors' are doubting presence of an opioid crisis
· Line 101: “…dyadic adjustment is one of the most important contributors to mental health” – If this is the basis of review, then more research support is needed. Define dyadic adjustment and why this is true.
o Strengthen section 1.3.
o Consider cutting parts of introduction that does not have bearings on this
· Lines 103-104: “…the influence is bidirectional” – citation?
· Line 119: ‘illness work’ – who authored this term? Is this a term utilized in literature?
· Line 124: “(if they weren’t working)” – unnecessary specification
· Lines 140-141: provide context of study
· Line 150: “and so sacrifice their personal needs” – consider rephrasing and provide more context
· Lines 154-157: Does more research support partner “disconnection or connection”?
· Lines 159-161: Provide more research on strength of partner relationships
· Line 169: “for example sacrificing their own emotional needs” – repetitive, this example was used in Line 150 with the same citation
· Line 260-265: run on sentence makes it difficult to understand, simplify and rephrase
· Line 267: “which will in turn negatively impact the PWCP” – will or can?, cannot assume causality
· Line 269: “role of attachment style” – where is the research on this? Where has this come into play?
o Addressed again in line 286, but should be addressed prior to future directions
· Line 288: “The model…” – what model?
· Line 294: “There is substantial evidence…” – author cited only one study, where is the evidence?
Reviewer 5 Report
Comments and Suggestions for Authors
This is a concise and well written paper.
The research explores how chronic pain in individuals affects the mental and physical well-being of their partners and highlights the importance of considering partners in pain management approaches.
The topic is relevant and fills a gap by focusing on the often-overlooked impact of chronic pain on partners. It highlights the need to include partners in treatment plans and considers the dyadic nature of coping with chronic pain.
The study adds a broader understanding of the reciprocal relationship between chronic pain and partner well-being. It emphasizes the importance of relational approaches and identifies areas for future research to improve outcomes for both individuals and their partners. Families having a person with chronic pain (PWCP) know very well the problem described in the manuscript, but often they do not consider the painful life of the partner of PWCP and this manuscript reveals that the influence of pain is "bidirectional".
The text in rows 208 and 221 is not clear (unlike the rest of the manuscript). It would be good to clearly state what the "recent SR" compared.
The authors could also provide more detail on how studies were selected and analyzed in the narrative review. Explaining the inclusion and exclusion criteria would strengthen the reliability of the findings.
The conclusions align with the evidence and effectively address the main question. Overall, very good job.
Round 2
Reviewer 1 Report
Comments and Suggestions for Authors
no comments, happy with responses to review
Author Response
Thank you
Reviewer 2 Report
Comments and Suggestions for Authors
I still do not agree that the marriage has such an important impact on partners committing to support each other. Its not proven. And if so, please add a reference. Nowadays, couples do not necessarily marry each other and still support each other. Also a lot of people are getting divorced.
Author Response
I still do not agree that the marriage has such an important impact on partners committing to support each other. Its not proven. And if so, please add a reference. Nowadays, couples do not necessarily marry each other and still support each other. Also a lot of people are getting divorced.
Response
There is no longer any reference to "marriage" or "wedding" in this article. The first sentence has been revised to
"An important element of primary romantic relationships is the idea that couples undertake to support each other, despite the challenges that they may face in life".
Reviewer 3 Report
Comments and Suggestions for Authors
Most of the reviewer's suggestions have been addressed.
Author Response
Thank you
Reviewer 4 Report
Comments and Suggestions for Authors
Overall, the authors addressed many of the concerns from the reviewers. A few lingering concerns remain.
· The use of contractions (don’t verses do not) decrease from the professionalism of the manuscript. Below are a few examples. Please change from contraction form into the full words. Line 58, Line 202, Line 311, Line 353
· The phrase “so-called opioid crisis” was not removed, and double quotation marks were not removed from the term “opioid crisis” – line 107. The use of “so called” diminishes the impact of the opioid crisis on patients and families. Please carefully review the entire manuscript for this phrasing and update.
· Please add the definition of ‘mutual maintenance’ cycle to the manuscript with a citation.
· Additional citations needed for Line 56-58.
· Please rephrase Lines 197-199, Lines 315-318, sentence structures make the conclusions unclear.
· Please add clarification of Lines 194-196, which was not addressed in the original resubmission. The authors wrote “Thus the effects of chronic pain on a couple’s relationship are highly dependent on the ways in which couples are able to negotiate the stresses of this condition together.” It would be helpful to expand on the variable associated with resilience/positive relationships verses risk/more difficult relationship experiences. Noting that a relationship is highly dependent on how the couple navigates stress together seems anecdotal and requires some additional research. Attachment theory was added; however, the information does not fully support all the factors that go into navigating the stressors of having a partner with pain.
· Lastly, perhaps adding in a final summary paragraph with a few key points could help highlight some next steps that researchers and clinicians could take when considering partners of patients with chronic pain.
Round 3
Reviewer 4 Report
Comments and Suggestions for Authors
The authors have addressed all comments.
Author Response
Thank you